# Learning multiple variable-speed sequences in striatum via cortical tutoring

James M Murray*, G Sean Escola

Center for Theoretical Neuroscience, Columbia University, New York, United States

**Abstract** Sparse, sequential patterns of neural activity have been observed in numerous brain areas during timekeeping and motor sequence tasks. Inspired by such observations, we construct a model of the striatum, an all-inhibitory circuit where sequential activity patterns are prominent, addressing the following key challenges: (i) obtaining control over temporal rescaling of the sequence speed, with the ability to generalize to new speeds; (ii) facilitating flexible expression of distinct sequences via selective activation, concatenation, and recycling of specific subsequences; and (iii) enabling the biologically plausible learning of sequences, consistent with the decoupling of learning and execution suggested by lesion studies showing that cortical circuits are necessary for learning, but that subcortical circuits are sufficient to drive learned behaviors. The same mechanisms that we describe can also be applied to circuits with both excitatory and inhibitory populations, and hence may underlie general features of sequential neural activity pattern generation in the brain.

## Introduction

Understanding the mechanisms by which neural circuits learn and generate the complex, dynamic patterns of activity that underlie behavior and cognition remains a fundamental goal of neuroscience. Of particular interest are sparse sequential activity patterns that are time-locked to behavior and have been observed experimentally in multiple brain areas including cortex (*Luczak et al., 2007*; *Jin et al., 2009*; *Harvey et al., 2012*), basal ganglia (*Jin et al., 2009*; *Rueda-Orozco and Robbe, 2015*; *Mello et al., 2015*; *Dhawale et al., 2015*; *Gouvêa et al., 2015*; *Bakhurin et al., 2017*), hippocampus (*Nádasdy et al., 1999*; *Pastalkova et al., 2008*; *MacDonald et al., 2013*; *Eichenbaum, 2014*), and songbird area HVC (*Hahnloser et al., 2002*; *Kozhevnikov and Fee, 2007*). These experiments reveal key features of the relationship to behavior that must be recapitulated in any circuit-level model of sequential neural activity. First, these sequences have been shown to temporally rescale (i.e., contract and dilate) as behavior speeds up and slows down (*Mello et al., 2015*; *Gouvêa et al., 2015*), while human psychophysics experiments have shown that after learning a behavior (e.g., an arm reach trajectory) at one speed of execution, it can be reliably expressed at other speeds without additional learning (*Goodbody and Wolpert, 1998*; *Joiner et al., 2011*; *Shmuelof et al., 2012*). Thus the neural circuits underlying such behaviors must be able to extrapolate from the sequential activity generated at a particular speed during learning to temporally rescaled patterns. Second, the same neural circuits exhibit the capacity to flexibly generate different sequential activity patterns for different behaviors (*Pastalkova et al., 2008*; *Harvey et al., 2012*; *MacDonald et al., 2013*). Third, the process of learning imposes additional constraints on models of sequence generating circuits beyond the need for biologically plausible learning rules. As discussed in greater detail below, lesion and inactivation experiments have been shown to prevent learning without impairing the expression of learned behaviors (*Miyachi et al., 1997*; *Kawai et al., 2015*) suggesting that the neural circuitry of learning and execution can be decoupled.

*For correspondence: jm4347@ columbia.edu

Competing interests: The authors declare that no competing interests exist.

These results indicate a need for a comprehensive model to study the generation of sparse, sequential neural activity. Such a model should facilitate temporal rescaling and extrapolation to new speeds, flexibly change in order to meet changing contexts or behavioral goals, and decouple learning from performance. Several mechanisms have been proposed for generating sparse sequences, for example continuous attractor models (*Rokni and Sompolinsky, 2012*), reservoir computing networks (*Rajan et al., 2016*), and feedforward excitatory architectures including synfire chains (*Abeles, 1991*; *Goldman, 2009*; *Fiete et al., 2010*; *Veliz-Cuba et al., 2015*). Each of these models suffers limitations, however. For example: attractor models require finely tuned synaptic connectivity and single-neuron tuning properties; trained recurrent neural networks lack biologically plausible learning rules and connectivity constraints; and synfire chains have limited ability to temporally rescale their dynamics. In this work we propose a model which allows for (i) arbitrary speeding up and slowing down of the activity pattern without relearning of the synaptic weights, (ii) the flexible expression of multiple sequential patterns, and (iii) an arbitrary sequence to be learned from a time-dependent external input and reproduced once that external input is removed.

Although the basic mechanisms that we propose may be realized in any brain area where sequences have been observed, for a specific example and to compare to experimental results, we focus on neural activity in striatum, where sparse activity sequences have been observed in recurrently connected populations of inhibitory medium spiny neurons (MSNs) in rodents during locomotion (*Rueda-Orozco and Robbe, 2015*) and lever-press delay tasks (*Mello et al., 2015*; *Dhawale et al., 2015*; *Gouvêa et al., 2015*). The striatum collects inputs from many areas of cortex and thalamus, plays an important part in controlling learned movements using its projections via the output structures of basal ganglia to motor thalamus and brainstem (*Parent, 1990*; *Grillner and Robertson, 2015*), and has a central role in reinforcement learning (*Graybiel, 2005*). MSNs, which constitute over 90% of the neurons in striatum (*Gerfen and Surmeier, 2011*), exhibit stereotyped sequential firing patterns during learned motor sequences and learned behaviors in which timing plays an important role, with sparse firing sequences providing a seemingly ideal representation for encoding time and providing a rich temporal basis that can be read out by downstream circuits to determine behavior (*Jin et al., 2009*; *Mello et al., 2015*; *Rueda-Orozco and Robbe, 2015*; *Dhawale et al., 2015*; *Gouvêa et al., 2015*; *Bakhurin et al., 2017*). Such neural activity has been shown in rodents to strongly correlate with time judgement in a fixed-interval lever-press task (*Gouvêa et al., 2015*), and with kinematic parameters such as the animal's position and speed in a task in which the animal was trained to remain on a treadmill for a particular length of time (*Rueda-Orozco and Robbe, 2015*). In addition, pharmacological attenuation of neural activity in sensorimotor striatum has been shown to impair such behavior (*Miyachi et al., 1997*; *Mello et al., 2015*; *Rueda-Orozco and Robbe, 2015*), suggesting that sequential firing patterns in striatum are likely to play a causal role in an animal's ability to perform timekeeping and learned motor sequences. While motor cortex is also known to play a major role in both the learning and directing of motor behaviors such as limb movements in both primates (*Fritsch and Hitzig, 1870*; *Georgopoulos et al., 1982*, *1986*; *Moran and Schwartz, 1999*; *Kakei et al., 1999*) and rodents (*Wise and Donoghue, 1986*; *Kleim et al., 1998*; *Whishaw, 2000*; *Harrison et al., 2012*), cortical lesion studies have long pointed to the ability of subcortical structures to direct a large repertoire of movements, particularly 'innate' movements and learned movements that don't require dexterous hand motion (*Lawrence and Kuypers, 1968*; *Sorenson and Ellison, 1970*; *Bjursten et al., 1976*; *Passingham et al., 1983*).

Importantly, sequences of neural activity in the striatum exhibit the ability to temporally rescale their dynamics, dilating or contracting the sequence duration by up to a factor of five in proportion to the time-delay interval for obtaining a reward (*Mello et al., 2015*). A successful model should therefore have the ability to dynamically rescale its activity, ideally with the ability to generalize to new speeds after training at a particular speed, and without requiring relearning of the synaptic weights each time there is switching to a slower or faster trial.

As in other brain areas, neural representations within striatum are specific to the behavior being performed and to the context in which it is executed (*Jin and Costa, 2010*; *Tecuapetla et al., 2014*). In constructing a model of sequential activity patterns, it is therefore important that neurons be able to participate in multiple sequences in a flexible, context-dependent manner. This might involve the selective activation, concatenation, and recycling of particular subsequences, as well as the capability of a circuit to switch between different operational modes, e.g. input-driven versus autonomously driven. All of these will be important features of the model that we construct below.

Addressing the roles of cortical and striatal circuits in the learning and performance of motor skills, one set of recent studies has shown that rats are unable to learn precisely timed lever-press sequences when motor cortex is lesioned, but are able to successfully perform the sequence if the lesion occurs after the learning has already taken place (*Kawai et al., 2015*; *Otchy et al., 2015*). It was therefore suggested that motor cortex may act as a 'tutor' to subcortical brain circuits during learning, and that these lower brain circuits eventually internalize the activity pattern, allowing them to drive behavior without receiving further instruction from the tutor once the behavior has been learned (*Kawai et al., 2015*). We therefore develop a model of such a tutor-to-student circuit, in which a cortical tutor teaches a particular activity sequence to striatum, which eventually becomes able to perform the sequence autonomously.

In this paper, we propose a model for how striatum could be structured in order to produce sequential activity patterns that can rescale to faster and slower speeds, flexibly select and combine particular subsequences specific to particular behaviors, and decouple learning from performance in a way consistent with the lesion experiments discussed above. A key element is the presence of synaptic depression at the inhibitory synapses between MSNs, which has been shown to exist experimentally (*Tecuapetla et al., 2007*) and which competes with the effect of feedforward excitatory input to determine the rate of switching of activity from one neuron cluster to the next. By adjusting the relative levels of these parameters, it is possible to dilate or contract the time dependence of neural activity sequences by an order of magnitude or more. Furthermore, we show that our striatal model can encode multiple sequences that can be expressed individually or pieced together into longer sequences as selected by the external input. Next, learning is addressed by introducing an anti-Hebbian plasticity rule at the synapses between MSNs, and we show how this enables the circuit to obtain the desired structure and internalize the dynamical activity pattern, so that temporally patterned input from cortex eventually becomes unnecessary as the behavior is learned. Finally, we show that the same mechanisms can be applied to circuits with both excitatory and inhibitory units, and hence may provide an explanation for the sequential firing patterns that have been observed in other brain areas including hippocampus (*Nádasdy et al., 1999*; *Pastalkova et al., 2008*; *MacDonald et al., 2013*; *Eichenbaum, 2014*) and cortex (*Luczak et al., 2007*; *Jin et al., 2009*; *Harvey et al., 2012*).

## Results

### Synaptic depression enables temporally controllable sparse activity sequences in striatum

Experimentally observed population activity patterns in striatum during learned behaviors are sparse and sequential, and these are the main features that we want our model network to exhibit in a robust manner. In order to achieve sparse activity, we make use of the well-known fact that recurrent inhibition can lead to a winner-take-all steady state in which a single unit or group of units (where a unit consists of a cluster of MSNs) becomes active and inhibits the other units in the network from becoming active. Indeed, recurrent inhibition is a hallmark feature of MSNs in striatum, and such a picture has previously been suggested to apply to striatum (*Wickens et al., 1991*; *Beiser and Houk, 1998*; *Fukai, 1999*). Although individual inhibitory synapses between MSNs are relatively sparse and weak on the scale of the currents needed to drive spiking in these neurons (*Jaeger et al., 1994*; *Tepper et al., 2004*), active populations of many MSNs firing together may more effectively mediate suppression between populations, in particular if these populations are also receiving sufficient background excitation from cortex and/or thalamus to keep them near the firing threshold (*Ponzi and Wickens, 2010*, *2013*; *Angulo-Garcia et al., 2016*), possibly in a metastable depolarized 'up state' (*Wilson, 1993*).

In addition to sparse activity, our model also requires a mechanism by which the activity can be made to switch from one unit to another, otherwise the network would lock into a single winner-take-all state. While other mathematically similar approaches are possible (see Appendix 1 for further discussion), in this paper we propose that this mechanism is short-term plasticity in the form of depressive adaptation at synapses between MSNs. Such synaptic depression has in fact been observed experimentally (*Tecuapetla et al., 2007*). The effect of synaptic depression is to weaken the amount of inhibition from an active unit onto inactive units over time. If all units also receive

constant external excitatory input, then eventually the inhibition may weaken sufficiently that the net input to one of the inactive units becomes positive, at which point the activity switches to this unit, and it begins to inhibit the other units. This competition between synaptic depression and the level of external input is the basic mechanism that determines the dynamics of activity switching. In particular, adjusting the level of external input can change the duration of time that it takes for activity to switch from one unit to the next, thus providing a mechanism for controlling the speed of an activity sequence in a robust manner without requiring any change in intrinsic properties of the neurons or temporally precise input to the network.

The dynamics of $x_i(t)$, which we think of as describing the activity level of a cluster of MSNs, and the associated synaptic depression factors $y_j(t)$ in our network model are described by the following equations:

$$\tau \frac{dx_i}{dt} = -x_i + \phi\left(\sum_{j=1}^{N} W_{ij} x_j y_j + x_i^{\text{in}}\right)$$

$$\tau_y \frac{dy_j}{dt} = -(y_j - 1)(1 - x_j) - (y_j - \beta)x_j. \tag{1}$$

The first equation describes the activity of unit $i$ as being determined by a nonlinear function acting on recurrent and external inputs. The recurrent synapses are inhibitory, with weights $W_{ij} \leq 0$, and the external input is excitatory, with $x_i^{\text{in}} \geq 0$. For concreteness, we take the nonlinear function to be the sigmoidal function $\phi(x) = 1/(1 + e^{-\lambda x})$, where $\lambda$ is a gain parameter. The second equation in (1) describes the dynamics of synaptic depression, where the dynamic variable $y_j(t)$ represents the depression of all outgoing synapses from unit $j$ with characteristic timescale $\tau_y$, which we take to be much greater than the membrane time constant $\tau$ (*Tecuapetla et al., 2007*). The first term on the right-hand side of the equation drives $y_j$ to attain a resting state value of 1 if the presynaptic unit $j$ is inactive, so that the synapse is fully potentiated. If the presynaptic unit becomes active, with $x_j \approx 1$, then the second term drives $y_j$ toward $\beta$, where $0 \leq \beta < 1$, so that the synaptic weight depotentiates to a finite minimum value when the presynaptic unit is active (*Tecuapetla et al., 2007*).

As described above and discussed in detail in Appendix 1, the model defined by (1) exhibits activity switching between units due to competition between the two terms in the argument of the nonlinear function $\phi(x)$. The second term is a positive external input, which tends to make $x_i$ active. The first term is a negative input from other units in the network, and becomes weaker over time as other units remain active due to decreasing synaptic weight $W_{ij} y_j(t)$. When the first term eventually becomes smaller than the second, the net input becomes positive, causing $x_i$ to become active and begin to inhibit other units.

In *Figure 1a–b*, we show a striatal model that is fully connected by inhibitory synapses, where all off-diagonal elements have the same inhibitory weight ($-1$) except for those connecting unit $j$ to unit $j + 1$, which are depotentiated by an amount $\eta$. This means that if unit $j$ is currently active, then unit $j + 1$ will become active next since it experiences the least amount of inhibition. *Figure 1c–d* show that the expected sequence of activity (which is repeating due to the fact that we also depotentiate the weight between the last and first units) is indeed obtained in such a network, and that the magnitude of the constant external input can be used to control the rate of switching. The period of the activity sequence slows down tremendously as $x^{\text{in}}$ approaches the synaptic depression parameter $\beta$. This slowing down allows for the temporal dynamics to be smoothly and reliably controlled, providing a potential mechanism consistent with recent experiments showing dramatic dilation of the time-dependence in population recordings of striatal neurons (*Mello et al., 2015*), without requiring new learning of the synaptic weights from one trial to the next. While an infinite range of dynamical scaling can be obtained in the idealized limit of $\tau/\tau_y \to 0$ and $\lambda \to \infty$, *Figure 1d* shows that attaining both very long and short switch times $T$ is possible even away from this idealized limit. Finally, *Figure 1e* shows that a substantial dynamical range of sequence speeds can be obtained even if control over the precise value of the input $x^{\text{in}}$ is limited, as may be the case due to noise in the system, preventing extremely slow and extremely fast sequence speeds.

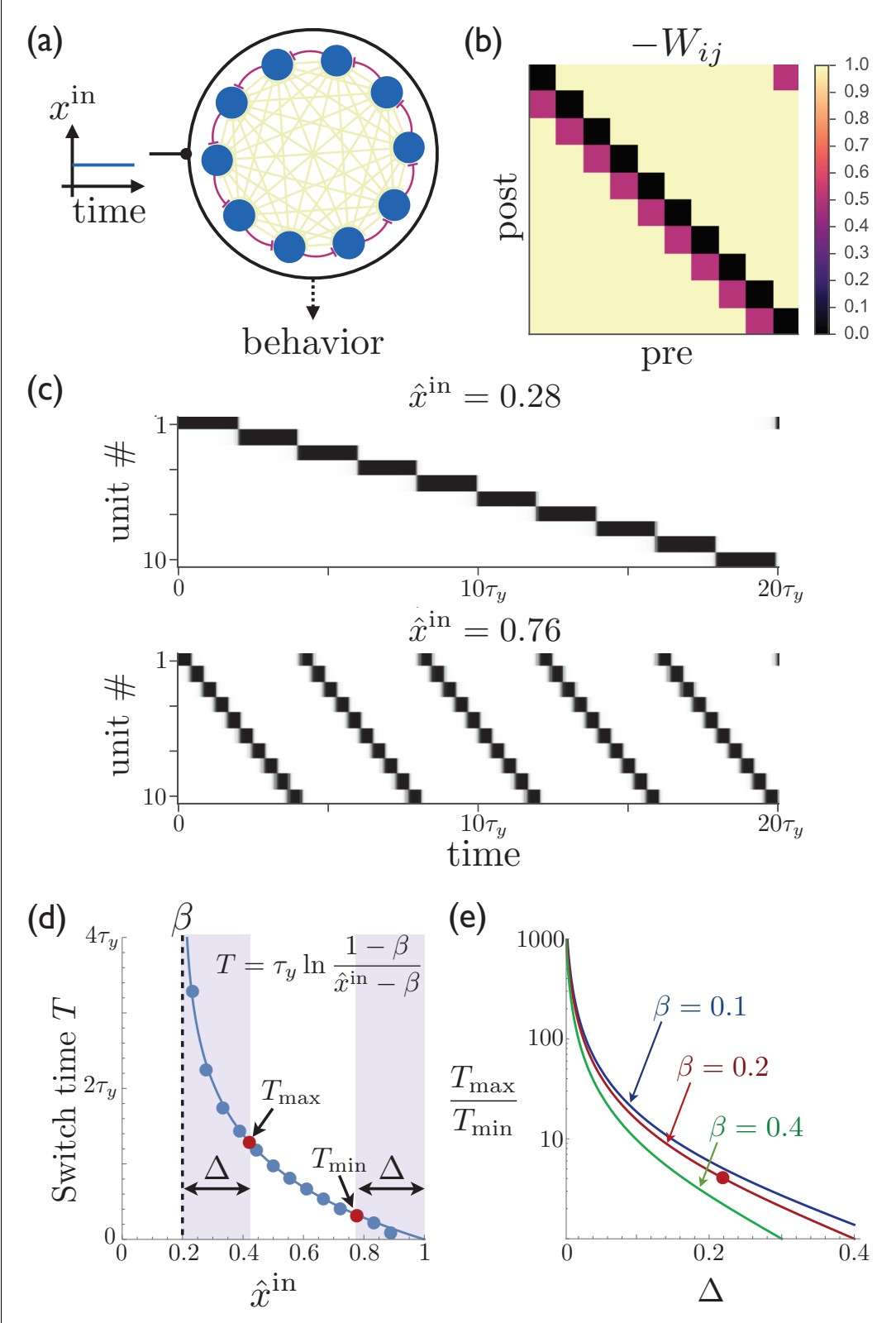

**Figure 1.** Rescalable sparse sequential activity in the striatum. (a) Schematic diagram of a 10-unit striatal network. Units receive constant external excitatory input and mutually inhibit each other. The burgundy synapses correspond to a depotentiated path through the network that enables sequential activity. (b) The synaptic weight matrix for the network shown in 'a', with subdiagonal weights depotentiated by $\eta = 0.2$. (c) The magnitude of the constant input $x^{\text{in}}$ to the network can be used to control the rate at which the activity switches from one population to the next. The units in the
*Figure 1 continued on next page*

*Figure 1 continued*

network are active in sequential order, with the speed of the sequence increasing as the excitatory input to the network is increased. Parameters for the synaptic depression are $\beta = 0.2$ and $\tau_y = 20\tau$, the gain parameter is $\lambda = 20$, synapses connecting sequentially active units are depotentiated by $\eta = 0.1$, and the effective input is $\hat{x}^{in} \equiv x^{in}/(1 - \eta)$. (d) The switch time as a function of the level input to the network. Points are determined by numerically solving *Equation 1*; curve is the theoretical result (equation shown in figure; see Appendix 1 for details). If the input is limited to the range $\beta + \Delta \leq \hat{x}^{in} \leq 1 - \Delta$ (e.g. because reliable functioning in the presence of noise would require the input to stay away from the boundaries within which switching occurs), then $T_{max}$ and $T_{min}$ are the maximum and minimum possible switching periods that can be obtained. (e) The temporal scaling factor is shown as a function of $\Delta$ for different values of the synaptic depression parameter $\beta$. The red dot corresponds to the ratio of the red dots in 'd'.

## Targeted external input selects which of several sequences striatum expresses

We can extend the model described so far to multiple—and even overlapping—behaviors by positing that the external input from cortex and/or thalamus targets the particular subset of MSNs needed to express a particular behavior. If multiple sequences are encoded in the weights between different populations of MSNs, then the external input can be thought of as a 'spotlight' that activates the behavior that is most appropriate in a particular context, with the details of that behavior encoded within the striatum itself, as shown in *Figure 2*. These subpopulations may even be partially overlapping, with the overlapping portions encoding redundant parts of the corresponding behaviors. In this way, a wide variety of motor behaviors could be encoded without requiring a completely distinct sequence for every possible behavior. This model dissociates the computations of the

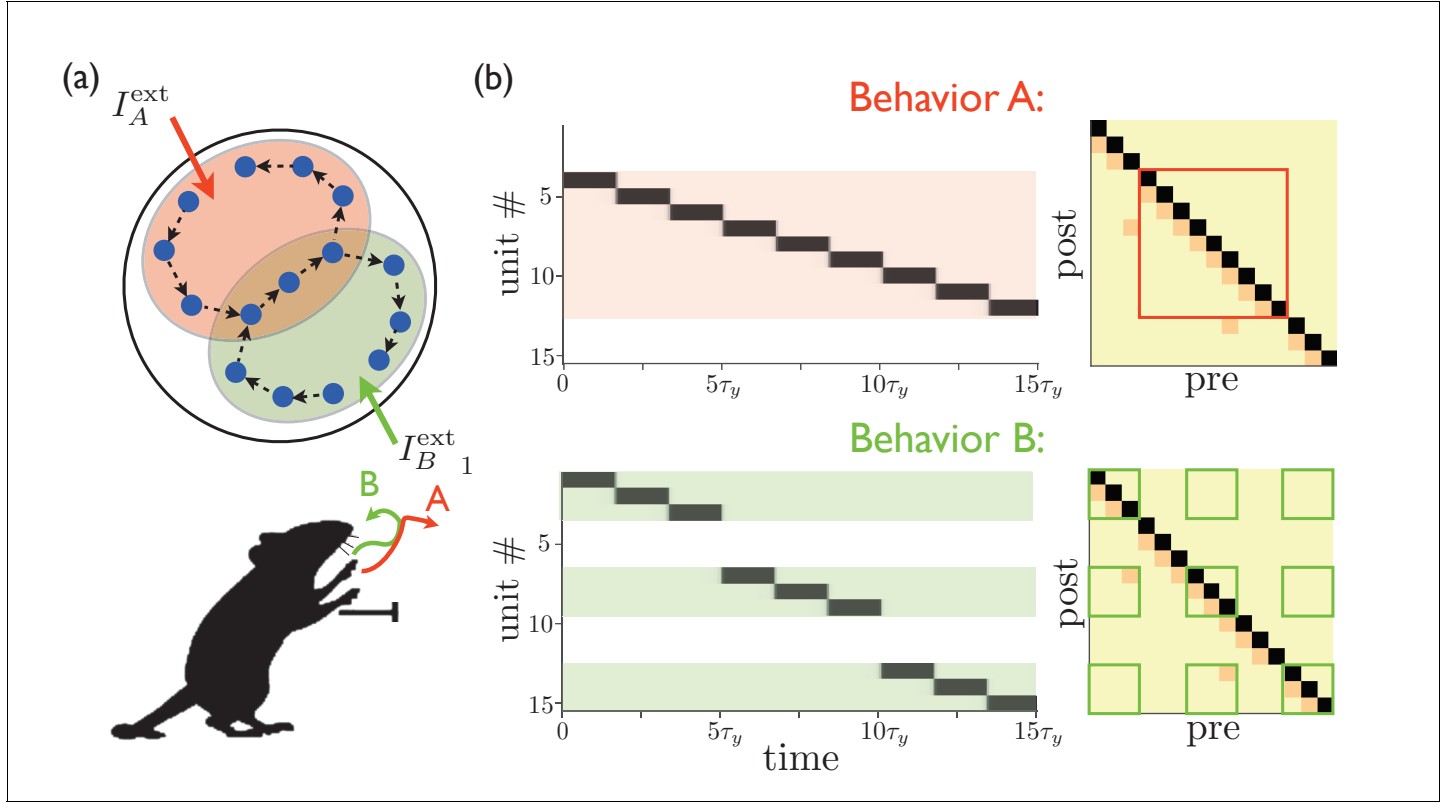

**Figure 2.** Targeted external input expresses one of several sequences. (a) Schematic illustration of partially overlapping striatal activity sequences selectively activated by external input. The arrows do not represent synaptic connections, but rather the sequence of activity within an assembly. Overlapping parts of the striatal activity sequences encode redundancies in portions of the corresponding behaviors (in this case, the middle portion of a paw movement trajectory). (b) Left panels show network activities in which only the shaded units receive external input. Right panels show the weights, with only the outlined weights being relevant for the network dynamics for each behavior.

selection and expression of motor sequence behaviors. The inputs to striatum select a sequence (possibly composed of several subsequences) by targeting a certain subpopulation of MSNs, and then the striatum converts this selection into a dynamical pattern of neural activity that expresses the behavior in time.

In addition to having multiple activity patterns being driven by different sets of tonically active inputs, the network is capable of operating in two distinct modes, as shown in *Figure 3*. In the first mode, the dynamics of neural activity is determined by the recurrent connectivity, as in the preceding examples, with external excitatory input providing a tonic 'go' signal to drive the network. In the second mode, on the other hand, the external input is strongly time-dependent and unique for each unit. Such input effectively overrides the pattern stored in the recurrent connectivity and enslaves the network dynamics to top-down input. In addition to providing an alternative route to controlling the network dynamics, such time-dependent external input can also play a role in facilitating learning, as we discuss below.

## Anti-Hebbian plasticity enables sequence learning

A striatal network with initially random connectivity can learn to produce sparse sequential activity patterns when driven by time-dependent cortical input. We again consider a network described by (1), but now with distinct time-dependent external inputs $x_i^{\text{in}}(t)$ to each unit $i$, and dynamic synaptic weights described by

$$\frac{dW_{ij}}{dt} = -\alpha_1 W_{ij} x_i \bar{x}_j - \alpha_2 (W_{ij} + 1)(1 - x_i)\bar{x}_j,$$

(2)

where $\bar{x}_j$ is the activity of unit $j$, low-pass filtered over a time scale $\tau_w$, and $\alpha_1$ and $\alpha_2$ control the rates of learning. Roughly, $\bar{x}_j(t)$ will be nonzero if unit $j$ has been recently active over the time window from $t - \tau_w$ to $t$. The first term in (2) thus causes $W_{ij} \to 0$ if postsynaptic unit $i$ is active together with or immediately following presynaptic unit $j$. Otherwise, if $j$ is active but $i$ is not active, the second term causes $W_{ij} \to -1$. (It is important that $\tau_w$ should not exceed the typical time for which a unit

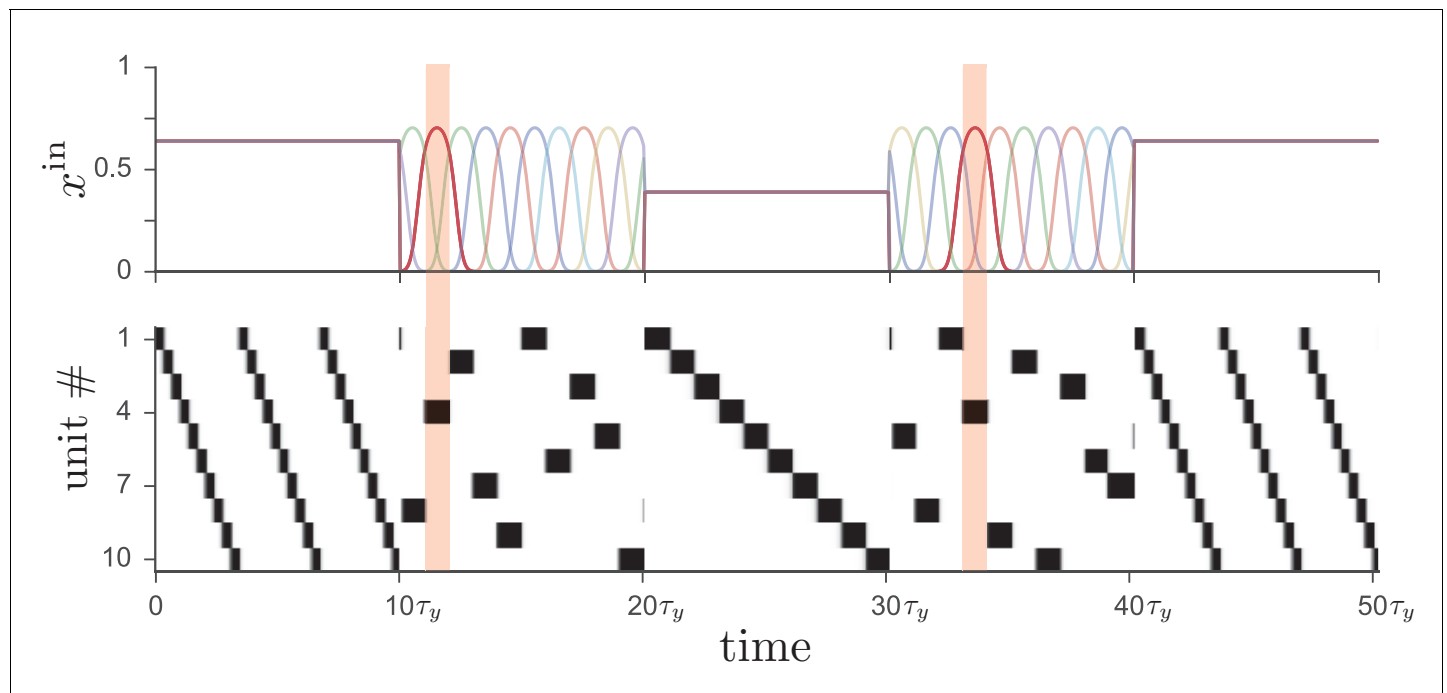

**Figure 3.** Driving the network from *Figure 1* with two different types of input leads to distinct operational modes. If input to the network (*top*) is tonic, the recurrent weights cause the network to produce a particular activity sequence (*bottom*), as in *Figure 1*, with the magnitude of the input controlling the speed of the sequence. Alternatively, if the input is strongly time-dependent, then the network dynamics are enslaved to the input pattern. In this case each unit receives a unique pulsed input, with the input to unit four highlighted for illustration.

remains active, or else there will be synaptic depotentiation not only between sequentially active units, but also between units separated by two or more steps in the sequence.) *Equation (2)* thus describes an anti-Hebbian learning rule, according to which synapses connecting units that fire together or in sequence are depotentiated, while others are potentiated.

*Figure 4* shows that sequences can develop in the network when it is subjected to several cycles of time-varying external input. *Figure 4a* shows that a network initially having no special features in its connectivity matrix can acquire such structure through anti-Hebbian plasticity by subjecting it to a repeated sequence of pulse-like inputs, which induce a regular pattern of sequential activity. It also shows that tonic input to the network leads to sparse activity sequences both before and after training, with short, random sequences occurring before training and long, specific sequences occurring after training. We have found that successful sequence learning can occur for a range of input sequence speeds when the pulse width is $\geq \tau_y$, but that some units in the sequence will be skipped if the input varies on time scales $\ll \tau_y$ (data not shown). In *Figure 4b*, the regular input to each unit is replaced by a superposition of sinusoids with random amplitudes and phase shifts. (We presume that in the brain cortical input to the striatum is structured in a meaningful way rather than random, determined by a reinforcement learning process that we are not modeling explicitly. Our use of random input here, however, illustrates that robust sequences emerge naturally in striatum even in the case where the input is not highly structured.) This input leads to a particular activity sequence in the network, with only one unit being active at a given time due to the inhibitory competition between units. Meanwhile, synaptic weights between sequentially active units are depotentiated by anti-Hebbian learning, eventually leading to a weight matrix (labeled '$t = 500\tau_y$' in *Figure 4b*) in which each unit that is active at some time during the sequence is described by a column with a single depotentiated entry, which corresponds to the next unit to become active in the sequence. *Figure 4c* illustrates that it is also possible to train a network that has previously learned one sequence to produce a new unrelated sequence. The evolution of the synaptic weights during the learning and relearning phases is illustrated in *Figure 4d*, where the number of training cycles required for learning the sequence is determined by the learning rates $\alpha_1$ and $\alpha_2$ in (2).

After the network has been trained in this way, it is able to reproduce the same pattern of activity even after the time-dependent input is replaced by a constant excitatory input to all units. This is similar to the network model studied in above, although now with the active units appearing in random order. *Figure 4e* shows that, as in the earlier network model, the level of external input can be used to control the speed of the activity sequence, with the dynamical range spanning more than an order of magnitude. Finally, *Figure 4f* shows that, again replacing the time-varying input with constant external input to all units, the activity pattern and sequence speed in this trained network are robust with respect to random perturbations of the weights $W_{ij}$. For comparison, we show in *Appendix 2—figure 1* that performance is severely degraded by comparable perturbations in a reservoir computing system.

Previous models have shown that neural activity sequences can emerge from initially unstructured networks of excitatory neurons via spike-timing-dependent plasticity (STDP) (*Fiete et al., 2010*; *Veliz-Cuba et al., 2015*; *Ravid Tannenbaum and Burak, 2016*). Compared with these earlier works, our model has the advantage of being able to dynamically adjust the speed of the activity sequence, as shown in *Figure 4e* (*cf.* however Refs. [*Veliz-Cuba et al., 2015*; *Pehlevan et al., 2015*; *Tristan et al., 2014*], where some temporal rescaling in activity patterns has been obtained using distinct mechanisms). In addition, our model does not require the assumption of heterosynaptic competition limiting the summed synaptic weights into and out of each unit, as in Ref. (*Fiete et al., 2010*).

Taken together, the above results show, within the context of a highly simplified network model, that time-varying input can lead to robust activity sequences, but that this input is no longer necessary once the circuit has internalized the sequence. Further, the speed of the dynamics can be adjusted using the overall level of external input to the network. Taken as a model of striatum, it therefore provides a possible explanation of the motor cortex lesion studies of Ref. (*Kawai et al., 2015*), as well as the variable-delay lever press experiments of Ref. (*Mello et al., 2015*).

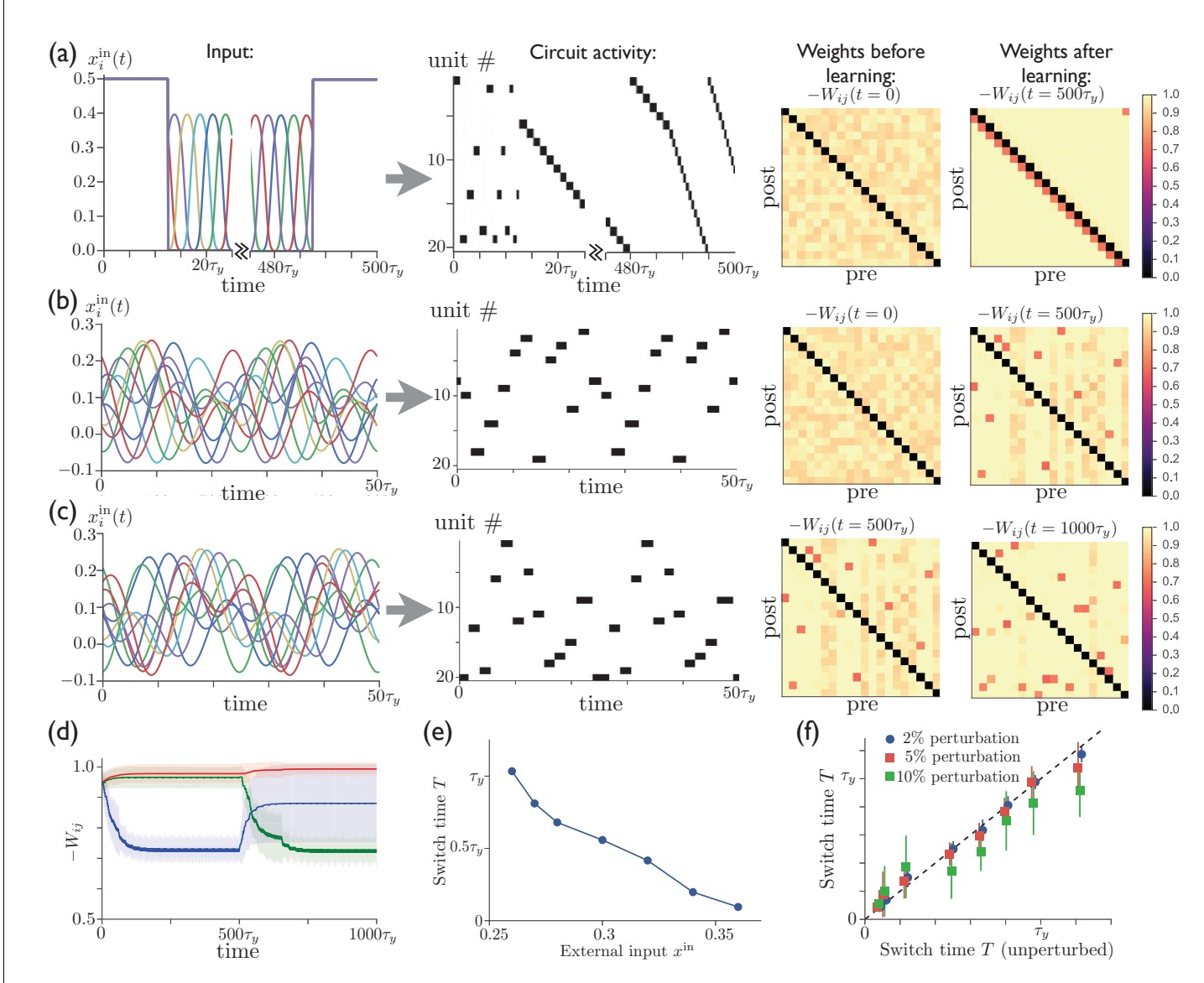

**Figure 4.** Repeated external input can tutor striatum to produce sparse activity sequences. (a) A regular sequence of pulse-like inputs (only 10 inputs are shown for clarity; $25\tau_y$ = 1 cycle) (*left*) leads to a sequential activity pattern (*center*) and, via anti-Hebbian plasticity, converts a random connectivity matrix (at $t = 0$) to a structured matrix after 20 cycles (at $t = 500\tau_y$) (*right*). The input during the first half of the first cycle and the last half of the last cycle has been replaced by constant input to all units, which leads to a short random sequence before training and to a long specific sequence following training, with the sequence speed determined by the level of tonic input. (b) Starting with random connectivity between units (at $t = 0$), each unit is driven with a distinct time-varying input consisting of a random superposition of sine waves (two cycles of which are shown for 10 inputs) which produces a repeating activity sequence. Anti-Hebbian learning results in a structured matrix after 20 cycles (at $t = 500\tau_y$). (c) 20 cycles with a new input elicits a different activity pattern and overwrites the prior connectivity to a new structured matrix (at $t = 1000\tau_y$). (d) The evolution of the synaptic weights during the learning in 'b' and 'c'. The blue, green, and red lines show the average weights of synapses involved in the first pattern, the second pattern, and neither pattern, respectively. (The weights shown in blue are not all repotentiated in the second training period due to the fact that some of these synapses are from units that are not active in the second sequence. For these weights $\bar{x}_j \approx 0$ in *Equation 2*, and thus they do not learn.) (e) The average time for switching from one unit to the next as a function of the constant external input after learning. (f) The switch times are robust to random perturbations of the weights. Starting with the final weights in 'c', each weight is perturbed by $\Delta W_{ij} = p\xi_{ij}\langle W_{ij}\rangle$, where $\xi_{ij} \sim \mathcal{N}(0, 1)$ is a normal random variable, and $p = 0.02, 0.05,$ or $0.10$. The perturbed switch times (slightly offset for visibility) are averaged over active units and realizations of the perturbation. Learning-related parameters are $\tau_w = 3\tau$, $\alpha_1 = 0.05/\tau$, and $\alpha_2 = 0.02/\tau$.

## A sparsely connected spiking model supports learning and execution of time-flexible sequences

*Figure 5* shows that, as in the continuous version of the model, temporally patterned input and recurrent plasticity can be used to train a recurrent inhibitory network having no initial structure in the recurrent weights to perform a particular firing sequence, with only one cluster of neurons active at any one time. Recent experimental work has indeed identified clusters of neurons in striatum that appear to function as transiently active cell assemblies (*Barbera et al., 2016*). Because we interpret the units studied in the continuous case above as clusters of neurons rather than individual neurons, full connectivity between units can be easily obtained even if connectivity between neurons is sparse, since some neurons in one cluster will always have synapses to some neurons in any other cluster if the clusters are sufficiently large. In *Figure 5*, the connection probability between all pairs of neurons is $p = 0.2$, showing that sparse connectivity between neurons is sufficient to enable one population to effectively inhibit another, as in the continuous model. Although we have adopted a simplified scheme in which each spiking neuron participates in only one cell assembly of concurrently active neurons, making the mapping from the neurons in the spiking model onto the units in the continuous model straightforward, the approach could be extended using the standard theory of attractor networks to allow for each spiking neuron to participate in multiple assemblies (see Ref. [*Curti et al., 2004*], as well as Ch. 17 of Ref. [*Gerstner et al., 2014*]). In addition, while the highly structured input used to train the network may at first appear highly artificial, we point out that similar sparse sequential activity patterns have been observed in motor cortex, which is a main input to

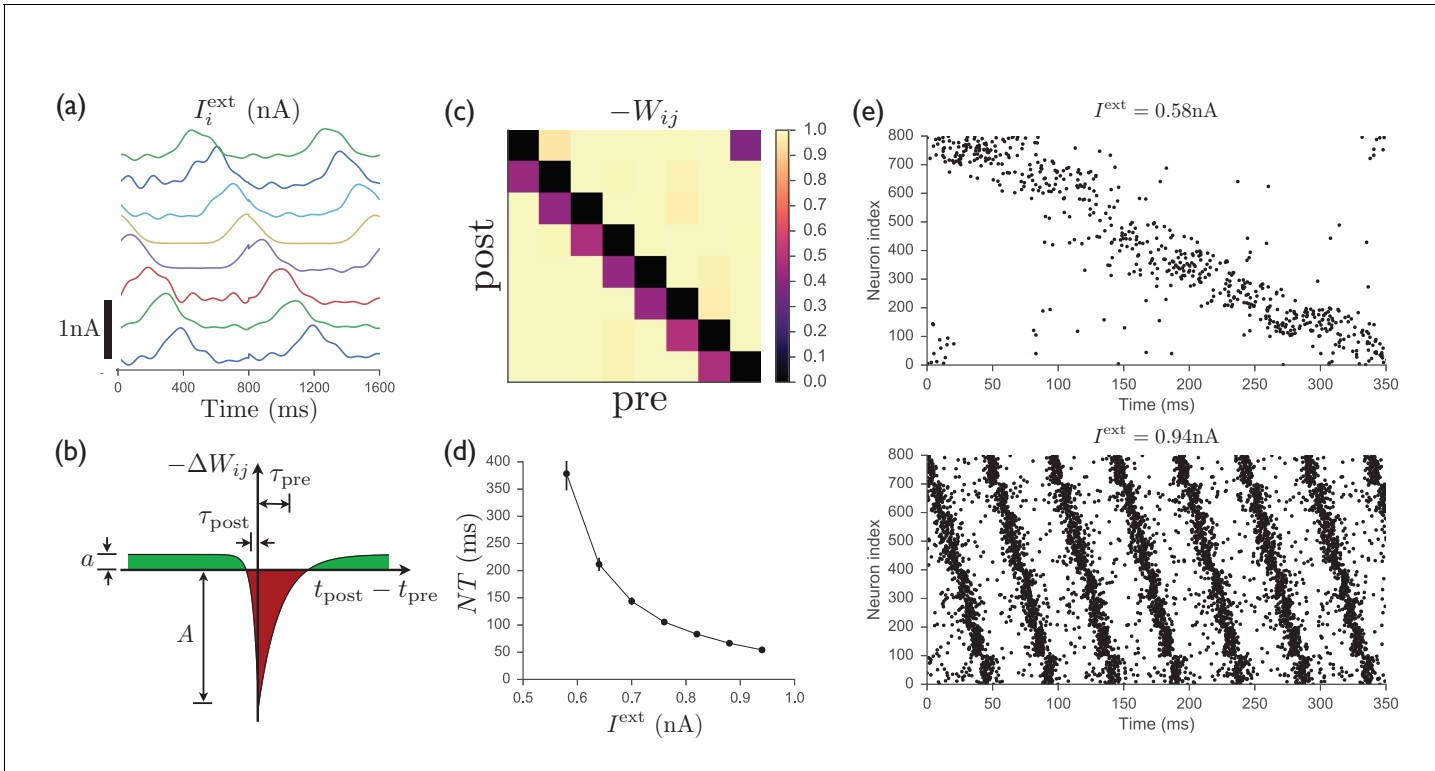

**Figure 5.** Spiking model of striatum. (a) Input currents to each of 8 distinct clusters of 100 neurons each (offset for clarity). This input pattern causes sequential activation of the clusters and is repeated noisily several times while the recurrent weights are learned. (b) Schematic anti-Hebbian spike-timing-dependent plasticity (STDP) rule for recurrent inhibitory synapses, showing that synapses are depotentiated when pre- and post-synaptic spikes are coincident or sequential, and potentiated if they are not. (This STDP curve applies whenever there is a presynaptic spike; there is no weight change in the absence of a presynaptic spike; see Materials and methods for specific mathematical details.) (c) Average recurrent inhibitory weights between clusters in a spiking network after learning with STDP. (d) After the weights have been learned, driving the network with tonic inputs of varying amplitudes leads to a rescaling of the period of the activity pattern. (e) Two examples of the time-rescaled activity patterns in the trained network with different values of tonic input current.

striatum, in rodents performing a learned lever-press task (*Peters et al., 2014*; *Dhawale et al., 2015*).

The STDP rule according to which recurrent inhibitory synapses are modified is shown in *Figure 5b*. The rule is anti-Hebbian, with postsynaptic spikes occurring at approximately the same time as or slightly after a presynaptic spike leading to weakening of the synapse, while presynaptic spikes occurring in isolation lead to a slight potentiation of the synapse. *Figure 5c* shows that this rule leads, after several repetitions of the input sequence, to a connectivity structure similar to that obtained in the continuous model, with decreased inhibition of a population onto itself and onto the next population in the sequence. Finally, as shown in *Figure 5d,e*, once the weights have been learned, constant input is sufficient to induce the desired firing pattern in the network, with the magnitude of this input controlling the rate at which the pattern progresses. Thus, as for the continuous network studied in above, the spiking network is able to learn a firing pattern from an external source, and later autonomously generate the same pattern over a wide range of speeds. Further details of the spiking model are presented in Materials and methods.

## Sparse sequential firing in an excitatory network with shared inhibition

Although, motivated by experimental results involving the basal ganglia, we have developed a model of recurrently connected inhibitory units, the same basic mechanisms for sequence learning can be applied to obtain sparse sequential firing patterns with flexible time encoding in a network of excitatory units with shared inhibition, a common motif used to obtain sparse coding of both static and dynamic neural activity patterns in models of cortical circuits. Such a network can be made to produce variable-speed sequential patterns just as in the circuit with direct lateral inhibition studied above. In this model, illustrated in *Figure 6a*, switching from from one excitatory unit to the next is again controlled by competition between the level of background input and synaptic depression at excitatory synapses, with the relative values of these quantities determining the rate at which activity jumps from one unit to the next. It is described by the following equations:

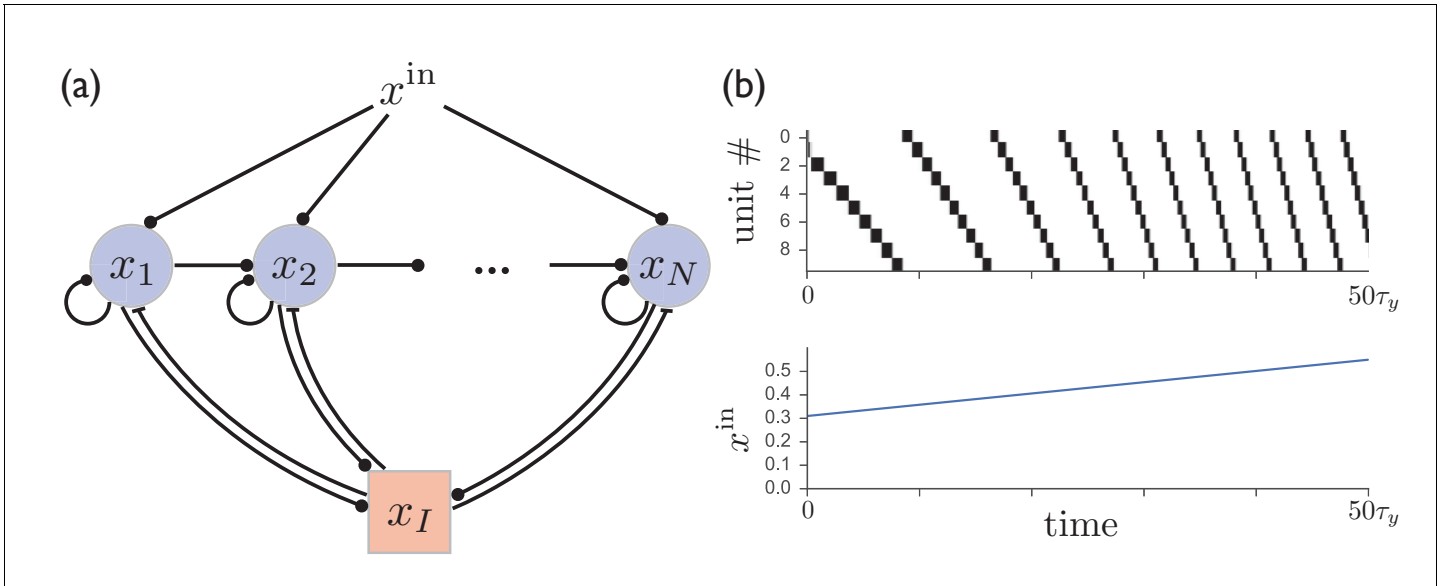

**Figure 6.** A network of excitatory units with shared inhibition exhibits variable-speed sequences. (a) A network of $N$ excitatory units are connected by self-excitation, feedforward excitation, and shared inhibition. (b) Just as in the recurrent inhibitory network, this network exhibits sparse sequential firing when the excitatory synapses are made depressing, with the speed of the sequence controlled by the level of external input. Parameters are $\beta = 0.2$, $\tau_y = 20\tau$, $\tau_I = \tau$, $J^{EI} = J^{IE} = 1$, $J_{ij} = 0.6\delta_{ij} + 0.2\delta_{j+1,j}$.

$$\tau \frac{dx_i}{dt} = -x_i + \phi \left( \sum_j J_{ij} x_j - J^{\mathrm{EI}} x_I + x_i^{\mathrm{in}} \right)$$

$$\tau_I \frac{dx_I}{dt} = -x_I + \phi_I \left( J^{\mathrm{IE}} \sum_j x_j \right), \tag{3}$$

where $x_i(t)$ is the activity of an excitatory unit, $x_I(t)$ is the activity of a shared inhibitory unit, and we assume $J_{ij}, J^{\mathrm{EI}}, J^{\mathrm{IE}} \geq 0$. In the case where the timescale characterizing inhibition is much faster than that characterizing excitation ($\tau_I/\tau \to 0$) and the nonlinearity of the transfer function for the inhibitory units can be ignored ($\phi_I(x) \approx x$), (3) becomes the following:

$$\frac{dx_i}{dt} = -x_i + \phi \left( -\sum_j (J^I - J_{ij}) x_j + x_i^{\mathrm{in}} \right), \tag{4}$$

where we have defined $J^I \equiv J^{\mathrm{EI}} J^{\mathrm{IE}}$. In the case where $J^I \geq J_{ij}$, and if excitatory synapses are made to be depressing by letting $x_j(t) \to x_j(t) y_j(t)$ on the right hand sides of the above equations, then this is precisely the model that was introduced in (1).

*Figure 6b* shows that the behavior of such a circuit with shared inhibition exhibits sparse sequential firing patterns virtually identical to those in the recurrent inhibitory network, even in the case where the above assumptions requiring inhibition to be fast and linear are relaxed by letting $\tau_I = \tau$ and $\phi_I(x) = \Theta(x) \tanh(x)$, where $\Theta(x)$ is the Heaviside step function. However, the dynamic range of temporal scaling factors that can be obtained in this case is somewhat more limited than in the model with direct lateral inhibition, with an approximately four-fold speed increase obtained in *Figure 6b*, compared with over an order of magnitude obtained in *Figure 1*. This is due at least in part to the finite timescale $\tau_I$ of the shared inhibitory population, which limits the speed at which activity switching can occur. This suggests that a purely inhibitory network, such as the the the one realized in striatum, may provide an advantage over an excitatory network with shared inhibition if involved in variable-speed tasks requiring a large dynamical range.

Although we shall not explore the effects of synaptic plasticity for this model in detail, the mapping of the network with shared inhibition onto the model previously studied, as shown in *Equation (4)*, means that sequence learning can also take place within this model. This requires that recurrent connections between excitatory synapses should follow a Hebbian plasticity rule, according to which synapse $J_{ij}$ is potentiated if unit $i$ fires immediately after unit $j$.

While models of sequence generation in networks containing both excitatory and inhibitory recurrent connections have been constructed previously, they generally have had limited or no control over sequence speed (*Fiete et al., 2010*; *Litwin-Kumar and Doiron, 2014*; *Pehlevan et al., 2015*; *Tully et al., 2016*). The ability to generate variable-speed sequential activity patterns may be relevant for various frontal and motor cortical areas, which can be described by the circuit architecture shown in *Figure 6a*, and which can exhibit sparse sequences similar to those in striatum during time-dependent decision making and motor-control tasks (*Harvey et al., 2012*; *Peters et al., 2014*; *Dhawale et al., 2015*). Given that these same cortical areas act as inputs to striatum, this mechanism for producing sparse sequences in circuits with shared inhibition may enable cortical networks to provide a pulse-like 'tutoring' input to striatum, as in *Figure 5*.

## Discussion

We have presented a model in which a network of recurrently connected inhibitory units internalize stereotyped sequential activity patterns based on temporally patterned input. Moreover, the same activity pattern can be reproduced after learning even after removing the temporally patterned input, and the speed of the activity pattern can be adjusted simply by varying the overall level of excitatory input, without requiring additional learning. As a model of striatum, we suggest that it may provide an explanation for recent experiments showing that (i) sparse sequences of neural activity in striatum dilate and contract in proportion to the delay interval in timekeeping tasks (*Mello et al., 2015*), and (ii) motor cortex is necessary to learn new behaviors but not to perform

already-learned behaviors (which are presumably directed at least in part by subcortical brain circuits such as the striatum) (*Kawai et al., 2015*).

In unlesioned animals, the ability to progress between two modes of control—one in which the dynamical neural activity in the basal ganglia is enslaved to top-down input from cortex, and another in which subcortical brain circuits generate dynamics autonomously—may form the basis of a cognitive strategy enabling performance of behaviors more reliably and with less cognitive effort as performance at a particular task becomes increasingly expert. As illustrated in *Figure 3*, the network can rapidly switch between these modes even after learning of a particular pattern has taken place, perhaps somewhat similar to the switching between the 'practice' and 'performance' modes of male songbirds in the absence or presence of a female, respectively (*Kao et al., 2005*). It is also well known that important differences exist in rodents, monkeys, and humans between goal-directed and habitual behaviors (for reviews, see Refs. [*Yin and Knowlton, 2006*; *Graybiel, 2008*; *Dolan and Dayan, 2013*; *Jahanshahi et al., 2015*]), and exploring the relation between these behavioral modes and the cortically and non-cortically driven modes described above is an important direction for future study. The question of where learned motor sequence memories are ultimately stored in the brain has been a subject of debate, with some studies favoring cortex—perhaps aided by 'tutoring' input from basal ganglia—as the ultimate storage site (*Desmurget and Turner, 2010*; *Hélie et al., 2015*), and others favoring subcortical structures such as sensorimotor striatum (*Miyachi et al., 1997*; *Kawai et al., 2015*). While our theory proposes a prominent role for striatum in storing motor sequence memories, it is also consistent with the possibility that motor sequence memories are effectively stored in *both* cortex and basal ganglia, so that inactivating either area in isolation will not necessarily abolish the learned behavior.

Assuming that the function of cortical input is to select from a set of possible learned behaviors as illustrated in *Figure 2*, we can ask what might be the potential role of corticostriatal plasticity, which many studies have shown is important for reward-based learning of motor behaviors in rodents, making these synapses a likely site for reinforcement learning (*Reynolds et al., 2001*; *Barnes et al., 2005*; *Yin et al., 2009*). If a behavior leads to a greater-than-expected reward, then a (possibly dopamine-mediated) feedback signal can cause the recently active corticostriatal synapses to be strengthened, making that behavior more likely to be performed in that particular context in the future, lowering the threshold for activation and possibly speeding up the activity sequence underlying a desired behavior, making the basal ganglia circuit important for controlling the 'vigor' associated with movements (*Yttri and Dudman, 2016*; *Dudman and Krakauer, 2016*). Although the model that we have developed focuses on plasticity within striatum rather than corticostriatal plasticity, it is worth emphasizing that it is consistent with both types of plasticity being present, with corticostriatal plasticity likely involved with action selection and degree of vigor, whereas MSN-to-MSN plasticity may be more important for encoding kinematic details of a given behavior. The scenario just outlined can be viewed as a generalization of recent models of reinforcement learning in mammals (*Fee, 2012*) to behaviors with temporally rich structure. Again using the spotlight analogy, it is also easy to see how multiple behaviors can be concatenated if the cortical and/or thalamic inputs activating the appropriate neuron assemblies are active together or in sequence. This provides a natural mechanism by which 'chunking' of simple behaviors into more complex behaviors might take place in the striatum (*Jin and Costa, 2010, 2015*).

Regarding the dynamical rescaling of neural firing patterns, several previous theoretical frameworks have been proposed for interval timing, including pacemaker-accumulators (*Gibbon, 1977*), in which a constant pacemaker signal is integrated until a threshold is reached; superposed neural oscillators (*Meck et al., 2008*), in which oscillations at different frequencies lead to constructive interference at regular intervals; and sequence-based models (*Killeen and Fetterman, 1988*; *Miller and Wang, 2006*; *Escola et al., 2009*), in which a network passes through a sequence of states over time. The last of these is most similar to the model that we present, though with the important difference that it involves stochastic rather than deterministic switching of activity from one unit to the next and hence has much greater trial-to-trial variability. In addition to these models, some previous theoretical works have attempted to use external input to control the speed of a 'moving bump' of neural activity within the framework of continuous attractors (*Burak and Fiete, 2009*; *Rokni and Sompolinsky, 2012*). However, in both of these previous studies, obtaining a moving activity bump requires external input that couples to different types of neurons in different ways.

This is not required by the model presented here, for which the activity bump still propagates even if the input to all units is identical.

It is also useful to contrast our model with other possible approaches within the framework of reservoir computing, starting with a random recurrent neural network (RNN) and training it to produce a sparse sequential pattern of activity either in the recurrent units themselves (*Rajan et al., 2016*) or in a group of readout units. Such training can be accomplished for example using recursive least squares learning (*Sussillo and Abbott, 2009*; *Laje and Buonomano, 2013*; *DePasquale et al., 2016*; *Goudar and Buonomano, 2017*) or various backpropagation-based algorithms (*Martens and Sutskever, 2011*). However, as we show in *Appendix 2—figure 1*, such a trained RNN generically tends to be much more sensitive to perturbations in the recurrent weights. In addition, the successful training of such an RNN requires many examples spanning the entire range of time scaling that one wishes to produce, whereas the network that we present can learn a sequence at one particular speed and then generalize to faster or slower speeds simply by changing one global parameter, making this network more flexible as well as more robust. This is reminiscent of the ability of human subjects learning a motor skill to successfully generalize to faster and slower speeds after training at a single fixed speed (*Shmuelof et al., 2012*).

We conclude by summarizing the experimental predictions suggested by our model. Central to the model is the anti-Hebbian plasticity rule that enables the inhibitory network to learn sequential patterns. Experimental results on medium spiny neurons in vitro have shown that recurrent synapses do in fact potentiate when presynaptic spiking is induced without postsynaptic spiking (*Rueda-Orozco et al., 2009*), as one would expect from the second term in (2). To our knowledge, however, the question of whether paired pre- and postsynaptic spiking would lead to depotentiation, as described by the first term in the equation, has not yet been addressed experimentally. Both Hebbian and anti-Hebbian forms of STDP at inhibitory synapses have been found in other brain areas, as reviewed in Ref. (*Vogels et al., 2013*). Because recurrently driven sequential activity of the sort that we describe requires depotentiation of inhibitory synapses between neurons both within the same cluster and from one cluster to the next sequentially active cluster, any activity-dependent learning rule must be qualitatively similar to the anti-Hebbian STDP curve shown in *Figure 5b*. The absence of such a learning rule would render it unlikely that learned recurrent connections within striatum play a major role in shaping learned sequences of neural activity, hence making this an important test of the theory.

Our model also predicts that the overall level of external excitatory input to the network should affect the speed of the animal's time judgement and/or behavior. By providing differing levels of input to a population of striatal MSNs optogenetically, it could be tested whether the speed of the neural activity sequence among these cells is affected. An alternative, and perhaps less technically challenging, approach would be to measure the overall activity level in the network, which should increase as the speed of the sequence increases. This effect should persist as long as saturation effects in activity levels do not become prominent (which does occur in the continuous model we present, but not in our spiking model). Changing the strength of recurrent inhibition should have a similar effect to changing the input level, although this would have to be done selectively to synapses between MSNs without disrupting feedforward inhibition from interneurons within striatum. Alternatively, dopamine may be able to cause a change of the sequence speed by modifying the synaptic depression parameter ($\beta$ in our model), and there is evidence from in vitro experiments that this indeed occurs (*Tecuapetla et al., 2007*). Thus changes in tonic dopamine levels should be able to effect temporal rescaling by modulating the input gain and/or recurrent synaptic depression, and indeed there has already been some evidence that such dopamine modulation occurs (*Soares et al., 2016*). However, it is as yet unknown whether direct- and indirect-pathway MSNs, which project to different targets within the basal ganglia (*Gerfen and Surmeier, 2011*), play distinct roles with regard to interval timing. Including both types of MSN in the model will be a natural extension for future work and will allow for more direct comparison with existing models of basal ganglia function (*Maia and Frank, 2011*; *Schroll and Hamker, 2013*).

Our theory also predicts that the neural activity pattern in striatum should be the same in trained animals before and after cortical lesions and that this neural activity should play a role in driving the animal's behavior. Investigating the neural activity in striatum and its role in generating behavior in lesioned animals would thus provide an important test of the theory. Observing the activity in cortex itself may also be useful. The theory suggests that time-dependent variability in cortical input is likely

to decrease as an animal becomes more expert at performing a task, or as it switches between behavioral modes. This could be studied via population recordings from striatum-projecting neurons in motor cortex.

Finally, while the lesion experiments of Ref. (*Kawai et al., 2015*) suggest that the instructive tutoring input to striatum likely originates in motor cortex, the source of the non-instructive input driving behavior and controlling speed after learning is unknown. It would be interesting for future experiments to explore whether the non-instructive input originates primarily from other cortical areas, or alternatively from thalamus, thereby endowing this structure with a functionally distinct role from cortex in driving behavior.

## Materials and methods

The following model describes a network of exponential integrate-and-fire neurons with synaptic depression used in *Figure 5*:

$$C\frac{dV_i}{dt} = g_L(E_L - V_i) + g_L\Delta_T \exp[(V_i - V_T)/\Delta_T] + I_i(t)$$
$$\frac{dx_{ij}}{dt} = \frac{1 - x_{ij}}{\tau_x} - ux_{ij}(t - 0^+)\sum_{t_j}\delta(t - t_j)$$
$$I_i(t) = I_i^{\text{ext}}(t) + uQ\sum_{j=1}^{N}x_{ij}(t)W_{ij}\sum_{t_j}\delta(t - t_j),\qquad(5)$$

where the membrane potential $V_i(t)$ is defined for each neuron $i$, and the dynamical synaptic depression variable $x_{ij}(t)$, which can be interpreted as the fraction of available neurotransmitter at a synapse, is defined for each synapse, with $x_{ij}(t - 0^+)$ meaning that the value of $x_{ij}$ just before the presynaptic spike should be used. When the membrane potential of neuron $i$ diverges, that is, $V_i(t) \to \infty$, a spike is emitted from neuron $i$, and the potential is reset to the resting potential $E_L$. Each time a presynaptic neuron $j$ fires a spike at time $t_j$, the depression variable is updated as $x_{ij} \to (1 - u)x_{ij}$, where $u$ is the fraction of neurotransmitter that is used up during each spike ($0 \le u \le 1$). The amount of electric charge that enters the postsynaptic cell during a presynaptic spike from neuron $j$ is $ux_{ij}(t)QW_{ij}$, where $Q$ has units of charge, and $u$, $x_{ij}$, and $W_{ij}$ are dimensionless. In terms of the original model described in the main text, each cluster of neurons corresponds to one of the units from the continuous model. As before, the competition between external input current and synaptic depression is used to obtain control over the temporal dynamics. The external inputs used in *Figure 5* are given by

$$I_i^{\text{ext}}(t) = x_0 + x_1 \sin^8\left(\frac{\pi t}{T} + \theta_i\right)\qquad(6)$$

where $x_0 = 0.2$ nA, $x_1 = 0.5$ nA, $T = 800$ ms, and $\theta_i = \pi n_i/8$ gives the phase shift of the input to each population $n_i = 1, \ldots, 8$. Noise, which was normally distributed and drawn independently for each cycle of the input, was added to $x_0$, $x_1$, and $\theta_i$, as well as by adding to (6) higher-frequency terms $\sim \sin(8\pi t/T)$ and $\sin(12\pi t/T)$ with random amplitudes and phase shifts. The other parameters used in *Figure 5* are $C = 300$ pF, $g_L = 30$ nS, $E_L = -70$ mV, $V_T = -50$ mV, $\Delta_T = 2$ mV, $\tau_x = 200$ ms, $u = 0.5$, $\tau_{\text{pre}} = 20$ ms, $\tau_{\text{post}} = 5$ ms, $Q = 1.5$ pC, $A = 0.05$, $a = 0.002$.

## Acknowledgements

The authors would like to thank L Abbott, S Kato, B Ölveczky, and J Seeley for helpful discussions and comments on the manuscript.

## Additional information

### Funding

| Funder | Grant reference number | Author |
| --- | --- | --- |
| National Institutes of Health | NIH DP5 OD019897 | G Sean Escola |

| Leon Levy Foundation | Fellowship | G Sean Escola |

The funders had no role in study design, data collection and interpretation, or the decision to submit the work for publication.

### Author contributions

JMM, GSE, Conceptualization, Formal analysis, Writing—original draft, Writing—review and editing

### Author ORCIDs

James M Murray, http://orcid.org/0000-0003-3706-4895

G Sean Escola, http://orcid.org/0000-0003-0645-1964

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

## Appendix 1

## Dynamics of model with synaptic depression

Intuition for the behavior of the model defined by (1) can be obtained by studying the case in which there are only two units, with activities $x_1(t)$ and $x_2(t)$, identical constant inputs $x_1^{\text{in}} = x_2^{\text{in}} = x^{\text{in}}$, and symmetric inhibitory connectivity $W_{ij} = \delta_{ij} - 1$. The behavior of this model is illustrated in **Appendix 1—figure 1**. This model can be understood by considering the fixed-point solutions for given fixed values of the synaptic depression variables $y_j$. In this case one can plot curves along which the time derivatives $\dot{x}_1$ and $\dot{x}_2$ vanish, as shown in **Appendix 1—figure 1**. The intersection of these curves describes a stable fixed point, which may occur at either $(x_1, x_2) \approx (1, 0)$ or $(x_1, x_2) \approx (0, 1)$, depending on which of $y_2$ or $y_1$ is larger. With this picture in mind we can now consider the effects of dynamical $y_j(t)$. Suppose that at a given moment $y_2 < y_1$ and hence $(x_1, x_2) \approx (1, 0)$. According to the second equation in (1), $y_2$ will begin increasing toward one due to the inactivation of $x_2$, while $y_1$ will begin decreasing toward $\beta$ due to the activation of $x_1$. As this happens, the net input to the second unit becomes positive, and the stable fixed point switches to $(x_1, x_2) \approx (0, 1)$ when $y_1 = x^{\text{in}}$ (assuming $\beta < x^{\text{in}} < 1$), and the synaptic depression variables begin adjusting to this new activity. The result will thus be repetitive switching between the two units being active, with the period of this switching determined by $\tau_y$, $\beta$, and (importantly) $x_{\text{in}}$. Versions of this two-unit model for switching, often termed a 'half-center oscillator,' have been previously studied in the context of binocular rivalry (**Seely and Chow, 2011**) and have long been used as a 'central pattern generator' in models of rhythmic behaviors (**Brown, 1914**; **Wang and Rinzel, 1992**; **Skinner et al., 1994**; **Marder and Bucher, 2001**).

The above analysis holds exactly in the limit $\tau/\tau_y \to 0$ and $\lambda \to \infty$, and in this limit it is straightforward to solve for the time that it takes for the activity to switch from one unit to the next:

$$T = \tau_y \ln\left(\frac{y_0 - \beta}{x^{\text{in}} - \beta}\right), \tag{7}$$

where $y_0 = \frac{1}{2}\left[1 + \beta + \sqrt{(1+\beta)^2 - 4x^{\text{in}}(1 + \beta - x^{\text{in}})}\right]$ is the largest value that $y_i(t)$ attains in each cycle and satisfies $x^{\text{in}} < y_0 \leq 1$. In a network with a large number of units, $y_0 \to 1$ since each unit has sufficient time to recover fully while other units in the network are active, and in this case (7) leads to the equation appearing in **Figure 1d** in the main text. **Equation (7)** shows that the switching period $T$ diverges logarithmically as $x^{\text{in}} \to \beta$ from above, and can be made arbitrarily small as $x^{\text{in}} \to y_0$ from below. Thus, in addition to allowing for neural activity to switch between populations, the competition between external input and synaptic depression also provides a mechanism for complete control of the speed of the network dynamics.

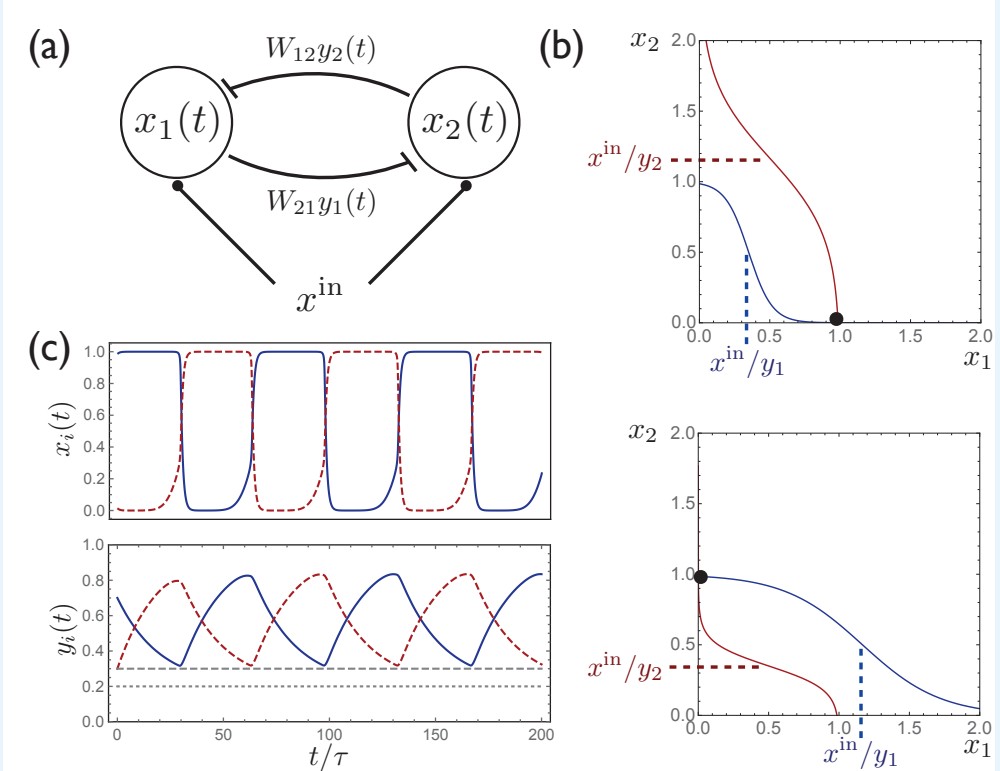

**Appendix 1—figure 1.** A simple circuit with two units exhibits activity switching. (**a**) A simple two-unit network with activities $x_1(t)$ and $x_2(t)$, symmetric inhibitory connectivity, and constant input $x^{\text{in}}$. (**b**) Curves along which, for given fixed values of $y_j$, the time derivatives $\dot{x}_{1,2} = 0$, with the intersection of these curves describing a stable fixed point. Depending on the relative values of $y_1$ and $y_2$, the fixed point occurs at either $(x_1, x_2) \approx (1, 0)$ (top) or $(x_1, x_2) \approx (0, 1)$ (bottom). (**c**) When $y_j(t)$ are included as dynamical variables, the synaptic depression leads to periodic switching between the two stable solutions.

Although we control temporal scaling throughout this paper by adjusting the external input level $x^{\text{in}}$, we note that essentially equivalent effects can be obtained within our model by instead adjusting the synaptic depression parameter $\beta$ rather than the external input $x^{\text{in}}$. While this might seem like an intrinsic neuron property that would be difficult to control externally, there is evidence from in vitro experiments that the degree of synaptic depression in MSNs in striatum is dependent upon the level of dopamine input to the neuron (**Tecuapetla et al., 2007**). What's more, changing dopamine levels in this circuit has been shown to reliably speed up or slow down an animal's time judgement (**Soares et al., 2016**), as one would expect from our model if the level of dopamine does in fact affect synaptic depression.

A possible objection to the above analysis is that $x^{\text{in}}$ cannot be tuned arbitrarily close to $\beta$ in the presence of noise, thus limiting the dynamical range of scaling parameters that can be obtained. In order to take this into account, we suppose that $\tilde{x}^{\text{in}} \equiv x^{\text{in}}/(1 - \eta)$ can only be tuned reliably to within precision $\Delta$. In this case, the maximum possible switching period that can be reliably obtained will no longer grow to infinity as $\tilde{x}^{\text{in}} \to \beta$, but rather will attain only a finite value as $\tilde{x}^{\text{in}} \to \beta + \Delta$. Similarly, the minimum attainable switching period cannot be arbitrarily small, but instead will reach a minimum value when $\tilde{x}^{\text{in}} \to 1 - \Delta$. Using (7), and taking $y_0 \to 1$, the dynamical range of temporal scaling is therefore given by

$$\frac{T_{\max}}{T_{\min}} = \frac{\ln\left(\frac{1-\beta}{\Delta}\right)}{\ln\left(\frac{1-\beta}{1-\beta-\Delta}\right)}.$$  (8)

*Figure 1c* shows that a large dynamical range can be obtained as a function of the noise parameter $\Delta$ even for biologically plausible noise values of $\Delta \geq 0.1$. Thus, an inhibitory network with synaptic depression and appropriately chosen synaptic weights is capable of performing an activity sequence over a wide dynamical range, even without requiring a biologically unrealistic degree of precision in the input to the network. Finally, we note that similar results to those shown in this section and in the main text can be obtained instead in a model which features depressive adaptation current rather than depressive synapses.

$$\tau \dot{x}_i = -x_i + \phi\left(\sum_j W_{ij} x_j - \gamma a_i + x_i^{\text{in}}\right)$$

$$\tau_a \dot{a}_i = -a_i + x_i,$$  (9)

where the depressive adaptation current $a_i(t)$, a low-pass filtered version of the activity $x_i(t)$, increases monotonically after unit $i$ becomes active, and $\gamma \geq 0$ is a constant describing the magnitude of the adaptation current. In this model, an active unit will tend to lower its own activity level over time due to the dynamical adaptation current $a_i(t)$. If this depression is sufficiently strong, then the unit may become inactive after some time, at which point another unit in the network will become active. As in the synaptic depression model studied in the main text, the switch time for successively active units can be dynamically adjusted by varying the level of external input $x^{\text{in}}$. Although this adaptation current model exhibits dynamics extremely similar to those of the synaptic depression model, we focus on the latter due to the fact that depressing synapses have been shown to be realized by neurons in the striatum (*Tecuapetla et al., 2007*).

## Appendix 2

### Time-interval scaling task in a trained random recurrent network

Traditionally, the motor cortex is viewed as the primary driver of voluntary motor output (**Fritsch and Hitzig, 1870**; **Evarts, 1968**; **Georgopoulos et al., 1986**; **Moran and Schwartz, 1999**; **Kakei et al., 1999**; **Churchland et al., 2006**; **Harrison et al., 2012**). Thus, as a point of comparison, we built a firing rate model of motor cortex with linear readout units representing striatal MSNs as schematized in **Appendix 2—figure 1a**. The cortical units in the model receive two inputs: one cueing the start of each trial and another cueing the target timing for the striatal pulse sequence on that trial. Of note, in contrast to the model presented in the main text, in this model striatum does not have any recurrent structure. The equations governing the model are as follows:

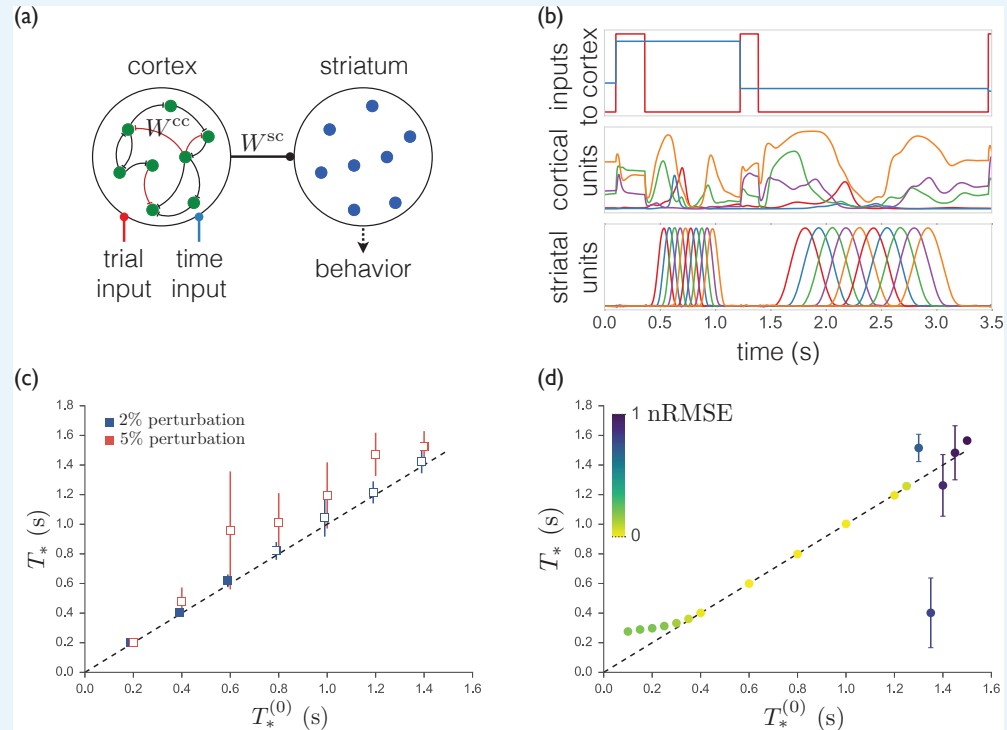

**Appendix 2—figure 1.** A trained recurrent network exhibits sequences that are less robust and less able to generalize. (**a**) Schematic of the model. The cortical units receive trial-specific inputs and project to striatum. Striatal units are not recurrently connected. The corticocortical and corticostriatal weights $W^{cc}$ and $W^{sc}$ are set as per the text. (**b**) Model simulation. *Upper*: Inputs to model. Red trace marks initiation of trials; blue trace indicates the target time for the trial. *Middle*: Sample cortical units. *Lower*: Striatal units. (**c**) Means and standard deviations of best-match times as a function of target times with random weight perturbations of 2 or 5%. Open symbols denote target times for which the nRMSE exceeded 0.3 on greater than 25% of trials. (**d**) Best-match times for a model trained on time intervals ranging from 0.4 s to 1.2 s, and then tested from 0.1 s to 1.5 s. The colors indicate the means of the nRMSEs of the trials at each target time.

$$\tau\dot{\mathbf{x}} = -\mathbf{x} + W^{cc}\mathbf{r}_\mathbf{c} + W^{cu}\mathbf{u}(t)$$
$$\mathbf{r}_\mathbf{c} = \tanh\mathbf{x}$$
$$\mathbf{r}_\mathbf{s} = W^{sc}\mathbf{r}_\mathbf{c},$$

where $\tau$ is the neuronal time constant, $\mathbf{u}(t)$ is the input at time $t$, $\mathbf{r}_\mathbf{c}$ and $\mathbf{r}_\mathbf{s}$ are the firing rates of the cortical and striatal units respectively, and $W^{cu}$, $W^{cc}$, and $W^{sc}$ are the input weights, recurrent corticocortical weights, and output corticostriatal weights respectively. We use a modified version of the FORCE algorithm (**Sussillo and Abbott, 2009**; **DePasquale et al., 2016**) to train $W^{cc}$ and $W^{sc}$ such that the duration of the pulse sequence of the striatal units matches the target time on each trial. **Appendix 2—figure 1b** shows the activity of the model after training, on two trials with different target durations.

Compared with the model presented in the main text, we find that our cortically driven model is (i) less robust to perturbations in the weights, and (ii) unable to extrapolate to perform the same sequence more quickly or slowly than it has learned in training. We measure the performance of the model in two ways. First, for each trial, we consider as templates all time-scalings of the pulse sequence in the range used to train the model (i.e., activity patterns such as those in the bottom panel of **Appendix 2—figure 1b**) and find the template with the best match to the produced striatal activity for that trial. The quality of the match is measured by the normalized root mean squared error: $\mathrm{nRMSE} = \sqrt{\left\langle ||\mathbf{r}_\mathbf{s}(t) - \hat{\mathbf{r}}_\mathbf{s}(t)||_2^2 \right\rangle / \left\langle ||\hat{\mathbf{r}}_\mathbf{s}(t)||_2^2 \right\rangle}$ where $\mathbf{r}_\mathbf{s}(t)$ and $\hat{\mathbf{r}}_\mathbf{s}(t)$ are the produced striatal activity and the template pulse sequence respectively. The best-match time is considered to be the response time of the model. Second, the value of the nRMSE indicates whether the response on that trial looked anything like a 'correct' striatal pulse sequence. By visual inspection, we set an nRMSE of 0.3 as the threshold above which a trial is not considered to be a meaningful pulse sequence.

In **Appendix 2—figure 1c** we show the mean and standard deviation of the best-match times for several target times after the addition of corticocortical synaptic weight noise. Notably, at 5% noise, the mean best-match times deviate far from the target times (compare to **Figure 4f**) and greater than 25% of the trials at every target time have nRMSEs exceeding 0.3.

We show the extrapolation performance of the model in **Appendix 2—figure 1d**. For target times shorter than the minimum target time used during training, the striatal responses deviate to longer times and the quality of the responses (as measured by the nRMSE) degrade. For target times longer than the maximum used during training, the responses quickly become meaningless with values of the nRMSE of about 1.

