## [Decision Letter]

Thank you for submitting your article "Learning multiple variable-speed sequences in striatum via cortical tutoring" for consideration by *eLife*. Your article has been favorably evaluated by Richard Ivry (Senior Editor) and two reviewers, one of whom, Michael Frank, is a member of our Board of Reviewing Editors. The following individual involved in review of your submission has agreed to reveal their identity: Mark D Humphries (Reviewer #1).

The reviewers have discussed the reviews with one another and the Reviewing Editor has drafted this decision to help you prepare a revised submission.

Summary:

The authors propose a model for how the striatum can generate sequences of neuron activity, of variable duration, from non-sequential input, and how they can emerge from tutoring from cortex, and can progress at varying speeds. Their mechanism rests on some form of adaptation – here picked to be synaptic depression at MSN-MSN synapses, given this has existing experimental evidence. With adaptation as the mechanism, dilation of temporal sequences can occur solely through changing the level of input drive. Learning such sequences is proposed to occur via anti-Hebbian plasticity at MSN-MSN synapses. Finally, the authors show that their results generalise to a sparse, spiking network model. Thus, the model makes sense of biophysical data (synaptic depression), shows how it can explain experimental data on population dynamics, and makes clear, testable experimental predictions.

Essential revisions:

Both reviewers were quite enthusiastic about your contribution, but just wanted to see you address the following points.

1) We would like to know a bit more about the dependencies of the model's behaviour on its parameters, particularly for learning. These can take the form of further text, or small simulations – we will defer to your preference here. Specifically, the tutoring (Figure 4) requires that the input sequence is periodically repeated; presumably the time-scale of the *period* of the sequence has to be >> *τ_y_*? What is the lower-limit here? In other words, how does the time-scale of interaction between neurons set by *τ_y_* dictate the fastest period of the tutoring sequence? Presumably this is also a function of *x*^in^, just as for the switch time in the constant input case to the static network (Figure 1).

2) Currently learning in the model is mediated by anti-Hebbian plasticity. The authors state in the Discussion that it still seems to be an open question as to whether this form of plasticity is appropriate for MSNs. Are there other learning rules that might also work for the model? Have the authors tried other learning rules? It might be nice to see more work or at least a discussion of potential alternative learning rules should the prediction of anti-Hebbian plasticity prove to be not well supported experimentally. This is not a criticism but rather something that is likely of interest to readers.

---

## [Author Response]

Essential revisions:

Both reviewers were quite enthusiastic about your contribution, but just wanted to see you address the following points.

1) We would like to know a bit more about the dependencies of the model's behaviour on its parameters, particularly for learning. These can take the form of further text, or small simulations – we will defer to your preference here. Specifically, the tutoring (Figure 4) requires that the input sequence is periodically repeated; presumably the time-scale of the period of the sequence has to be >> τ_y_? What is the lower-limit here? In other words, how does the time-scale of interaction between neurons set by τ_y_ dictate the fastest period of the tutoring sequence? Presumably this is also a function of x^in^, just as for the switch time in the constant input case to the static network (Figure 1).

The reviewers are certainly correct that there is a limit to how fast the tutoring input can become before learning can no longer take place. This can be understood most easily for regular, pulsatile inputs as in Figure 4, where the relevant parameter is T_in/*τ_y_*, where T_in is the duration of each input pulse (making NT_in the total period of the input). In the figure we have taken T_in/*τ_y_* = 1.25, so that each unit is active for a time ~*τ_y_* during training. If the speed of the input is increased so that T_in<<*τ_y_*, then sequence learning can still take place, but some of the units will be skipped, e.g. with only every other unit in the sequence becoming active. In practice, we find that this happens around T_in ~ τ_y_/2, though the precise value depends on other parameters such as the input amplitude. In order to make this point clear, we have added the following note to the text in the manuscript where Figure 4 is discussed: “We note that sequence learning can occur for a range of input sequence speeds, but that some units in the sequence will be skipped if the input varies on time scales << *τ_y_*.”

Regarding the other parameters in the model, our general approach has been to develop the simplest possible model that exhibits the effects we are interested while still following known biological constraints. The advantage of this approach is that much of the circuit’s behavior can be understood directly from analyzing the equations describing it, thus eliminating the need to test the effects of varying the circuit’s parameters in simulations. In general, we have endeavored to use what we judged to be reasonable values, to include some discussion about which values are permissible and why, and to check that the system’s qualitative behavior doesn’t depend on fine tuning. Below we summarize the most important of these choices, point out where we have already discussed them in the manuscript, and in some cases point out the modifications we have made to the text in order to further clarify the roles of these parameters:

*τ_y_/τ*: The ratio of the synaptic depression time constant to the membrane potential time constant must satisfy *τ_y_/τ* >> 1, and we have added a mention of this requirement below Equation (Abeles, 1991) in the main text. As we make clear from the theoretical arguments in Appendix A, the particular value of the ratio has very little effect on the dynamics of the model as long as this condition is satisfied. Because the membrane time constant of actual neurons is typically ~10ms while characteristic time scales for synaptic adaptation are >100ms, we are confident that the assumption used in our model is reasonable.

β: The allowable range of the synaptic depression parameter is 0 < β < 1. Smaller values of β are slightly preferable to larger ones since they increase the dynamic range of allowable inputs (see Figure 1). As we have shown in the simplified model in Appendix A, however, the value of β does not have a very significant effect on the network dynamics, hence we have fixed it to the experimentally plausible value of β=0.2 (see Tecuapetla et al., 2007) and have not varied it further in our simulations.

*α*_1_, *α*_2_ and *τ_w_*: The time scale of the eligibility trace for learning, *τ_w_*, should not exceed T, the typical time for which a unit remains active, or else there will be synaptic depotentiation not only between sequentially active units, but also between units separated by two or more steps in the sequence. The learning rates for synaptic depotentiation and potentiation, *α*_1_ and *α*_2_, set the characteristic time scale for learning, as shown in Figure 4. Changing the magnitude of these parameters will affect the minimum number of training cycles required for the system to learn a sequence pattern, and changing the ratio between them will affect the asymptotic values of the depotentiated synaptic weights, but such changes will not otherwise change the results qualitatively in any way. We have added text to the first and second paragraphs of the section on learning that clarifies the roles of and constraints on these three learning-related parameters.

2) Currently learning in the model is mediated by anti-Hebbian plasticity. The authors state in the Discussion that it still seems to be an open question as to whether this form of plasticity is appropriate for MSNs. Are there other learning rules that might also work for the model? Have the authors tried other learning rules? It might be nice to see more work or at least a discussion of potential alternative learning rules should the prediction of anti-Hebbian plasticity prove to be not well supported experimentally. This is not a criticism but rather something that is likely of interest to readers.

The reviewers’ question about the anti-Hebbian learning rule in our theory is an important one. In order to obtain activity sequences in a recurrent inhibitory circuit in the manner that we describe, the following conditions should be met: (i) synapses connecting neurons within a cluster should be depotentiated, so that they are able to fire together; (ii) synapses connecting neurons in sequentially active clusters (i.e. from cluster j to cluster j+1) should be depotentiated, so that they will activate in the correct order once the structured input is taken away; (iii) all other synapses should be potentiated, so that the clusters generally mutually inhibit one another. In order for these conditions to be met, the spike-timing-dependent plasticity rule must qualitatively resemble the one shown in Figure 5).

This can be made precise by noting that, averaged over the activity history of the network, changes in synaptic weights are given by dW_ij_ = \int F(t) C_ij(t) dt, where F(t) is the STDP curve, C_ij(t) is the time-dependent cross-correlation function of the activities of neurons i and j, and \int represents integration over time (see Tannenbaum and Burak, PLoS Comp. Bio. 2016). If the network activity is enslaved to the input during training, then C_ij(t) is effectively just the matrix of input correlations, and the three conditions above mean that F(t) should be positive at t=0 and for small positive t, and negative otherwise, as shown in Figure 5) (note that we’ve actually plotted the negative of this function).

Based on this, we conclude that an anti-Hebbian learning rule between MSNs in striatum is actually a strong requirement/prediction of our model. While the presence of such a learning rule would not unambiguously prove that recurrent connectivity in striatum plays a major role in driving sequences, the absence of such a learning rule would be very strong evidence against this hypothesis. In order to underscore this point, we have added the following text to the 7th paragraph of the Discussion section: “Because recurrently driven sequential activity of the sort that we describe requires depotentiation of inhibitory synapses between neurons both within the same cluster and from one cluster to the next sequentially active cluster, any activity-dependent learning rule must be qualitatively similar to the anti-Hebbian STDP curve shown in Figure 5). The absence of such a learning rule would render it unlikely that learned recurrent connections within striatum play a major role in shaping learned sequences of neural activity, hence making this an important test of the theory.”